# Multiparametric quantitative phase imaging for real-time, single cell, drug screening in breast cancer

Edward R. Polanco [1], Tarek E. Moustafa [1], Andrew Butterfield[2,3], Sandra D. Scherer [2,3], Emilio Cortes-Sanchez[2,3], Tyler Bodily [1], Benjamin T. Spike[2,3], Bryan E. Welm [2,4], Philip S. Bernard[2,5,6] & Thomas A. Zangle [1,2 ✉]

Quantitative phase imaging (QPI) measures the growth rate of individual cells by quantifying changes in mass versus time. Here, we use the breast cancer cell lines MCF-7, BT-474, and MDA-MB-231 to validate QPI as a multiparametric approach for determining response to single-agent therapies. Our method allows for rapid determination of drug sensitivity, cytotoxicity, heterogeneity, and time of response for up to 100,000 individual cells or small clusters in a single experiment. We find that QPI $EC_{50}$ values are concordant with CellTiter-Glo (CTG), a gold standard metabolic endpoint assay. In addition, we apply multiparametric QPI to characterize cytostatic/cytotoxic and rapid/slow responses and track the emergence of resistant subpopulations. Thus, QPI reveals dynamic changes in response heterogeneity in addition to average population responses, a key advantage over endpoint viability or metabolic assays. Overall, multiparametric QPI reveals a rich picture of cell growth by capturing the dynamics of single-cell responses to candidate therapies.

[1] Department of Chemical Engineering, University of Utah, Salt Lake City, UT, USA. [2] Huntsman Cancer Institute, University of Utah, Salt Lake City, UT, USA. [3] Department of Oncological Sciences, University of Utah, Salt Lake City, UT, USA. [4] Department of Surgery, University of Utah, Salt Lake City, UT, USA. [5] Department of Pathology, University of Utah, Salt Lake City, UT, USA. [6] ARUP Institute for Clinical and Experimental Pathology, Salt Lake City, UT, USA. ✉email: tzangle@chemeng.utah.edu

Precision oncology holds the promise of improving outcomes in cancer patients by tailoring effective therapies to an individual's tumor while minimizing toxic side effects from ineffective drugs[1]. Biomarker-driven personalized cancer treatment has been shown to improve response rates and extend progression-free survival[2]. Sequencing studies using large oncogene panels in advanced cancers find an actionable DNA mutation in 5–35% of cases, depending on associated tumor histology[3,4]. Although there are exceptional responders to targeted therapy[5,6], rarely do advanced cancer patients with a candidate "targetable" mutation exhibit long-term survival. Thus, there is a movement in precision oncology to implement functional cell-based assays to complement genomic panels[7].

Recent advances in tumor cell expansion have allowed for the development of ex vivo patient-derived models of cancer that faithfully recapitulate clinical behavior in terms of drug response[8–10]. Ex vivo testing is also amenable to clinical testing since it can be multiplexed and completed within weeks of tumor sample collection, allowing many more drugs to be screened at a lower cost, and on a timescale with the potential for informing patient care[11]. A variety of analytic methods for measuring the response of cultured cells to drug exposure are presently employed[10,12–14]. Cell culture-based drug-screening assays vary from simple cell counts and determination of live:dead ratios with stains, to metabolic assays (e.g., release of ATP or lactate[12]), to measurement of specific programmed cell death effectors such as caspases or BH3-domain activation[13]. CellTiter-Glo (CTG), for example, is an assay that measures cell ATP content as a proxy for cell viability. When used as an endpoint assay post drug exposure, CTG has been shown to produce reproducible[15] drug response data, more rapidly and with less bias than cell counting[16] and with greater signal-to-noise ratio than other luminescence assays such as Toxilight and resazurin-based assays[17]. However, these measures are typically applied as bulk, endpoint assays, and are incapable of capturing the dynamics of single-cell responses to therapy.

In contrast to endpoint assays, real-time assays can elucidate the temporal dynamics of drug response, and can discriminate between a cytostatic response where cell growth is substantially reduced and a cytotoxic response where the therapy induces cell death[18]. For example, incubator-housed microscope systems for measuring real-time cell proliferation (e.g., Incucyte) have been shown to yield results concordant to CellTiter-Glo and BH3 profiling[18–20]. As a longitudinal imaging approach, the Incucyte measures parameters such as population-averaged proliferation rate and cell viability throughout the experimental duration granting insight into changes in cell behavior throughout the course of imaging. An emerging alternative is to use cell mass accumulation rate as a measure of cell growth[21–23]. For example, suspended microchannel resonators are a highly sensitive tool for measuring changes in cell mass. Microchannel resonators can measure statistically meaningful changes in cell growth from very short duration (~10 min) measurements[24], or individual resonators can be used for longitudinal imaging of cell behavior in response to drugs[25]. However, this approach is limited by the need to flow cells through individual resonators and works best with non-adherent cell types.

Quantitative phase imaging (QPI) is a real-time, label-free technique for determining the growth of individual cells by measuring the phase shift of light as it passes through a transparent sample such as a cell[23,26]. This quantity is directly proportional to cell mass, which increases due to cell growth, such as during progression through the cell cycle[27,28]. QPI is a real-time, high throughput tool for measuring the growth response of individual cells to therapy. Previous applications of QPI have utilized microfluidics to study cell response to fluidic shear stress[29] or identify resistant cell populations using up to 20 different QPI-derived features such as area and shape[30]. These and other studies of QPI have narrowly focused on measuring the overall sensitivity[31] or toxicity[32] of potential therapies with only limited studies of the heterogeneity of response[33]. Previous work with CTG has shown that combining multiple measures of cell response is superior to only measuring drug sensitivity[34]. QPI, therefore, can have greater impact as a tool for functional medicine by enabling simultaneous measurement of multiple parameters that are indicative of cancer cell response to therapy.

Here we introduce QPI as a quantitative multiparametric method to characterize dynamic changes in growth rate, drug sensitivity, drug toxicity, heterogeneity, and time of response (ToR) with sufficient throughput to make this approach suitable for clinical applications. We demonstrate that these parameters, summarized in Table 1, are orthogonal measurements that cannot be derived from traditional measurements such as mean drug sensitivity alone. These parameters can also be combined to quantify drug-dependent, dynamic responses of cell populations in terms of time-varying mean and standard deviation of growth rates. Taken together, the QPI-derived parameters we develop here give a richer, more complete description of cell response to therapy than conventional drug screening approaches.

## Results

**Measurement of specific growth rate from QPI data**. We imaged breast cancer cell lines, representing diverse clinical subtypes (Table S1) using a custom QPI microscope (Fig. S1a, b) based on differential phase contrast (DPC) microscopy and phase reconstruction[35,36]. We chose DPC for image acquisition and phase reconstruction since its flexible design allows it to be customized for high throughput phase measurements. Additionally, the compact design of the final system fits inside a tissue culture incubator and requires only inexpensive optical components that are readily available (Table S2) facilitating widespread implementation as a clinical screening tool. Before imaging, we calibrated the system alignment and confirmed accuracy and precision by measuring the refractive index of polystyrene microbeads (Fig. S2). Our system is

**Table 1 Summary of QPI drug response parameters.**

| Name | Abbreviation | Description | Units |
|---|---|---|---|
| Specific growth rate | SGR | Exponential growth constant, computed as the rate of change of cell mass over time, normalized by cell mass | $h^{-1}$ |
| Half maximal effective concentration | $EC_{50}$ | Therapy concentration at which cells exhibit 50% of maximum response | µM |
| Depth of response | DoR | Maximum difference in SGR between cells at minimum and maximum therapy concentrations, computed from Hill sigmoid curve fitting parameters | - |
| Time of response | ToR | The average time required to elicit a response to therapy at the tested concentration | h |
| Standard deviation of response | SD | Standard deviations of SGR at the tested concentration as a measure of cell-to-cell heterogeneity | $h^{-1}$ |

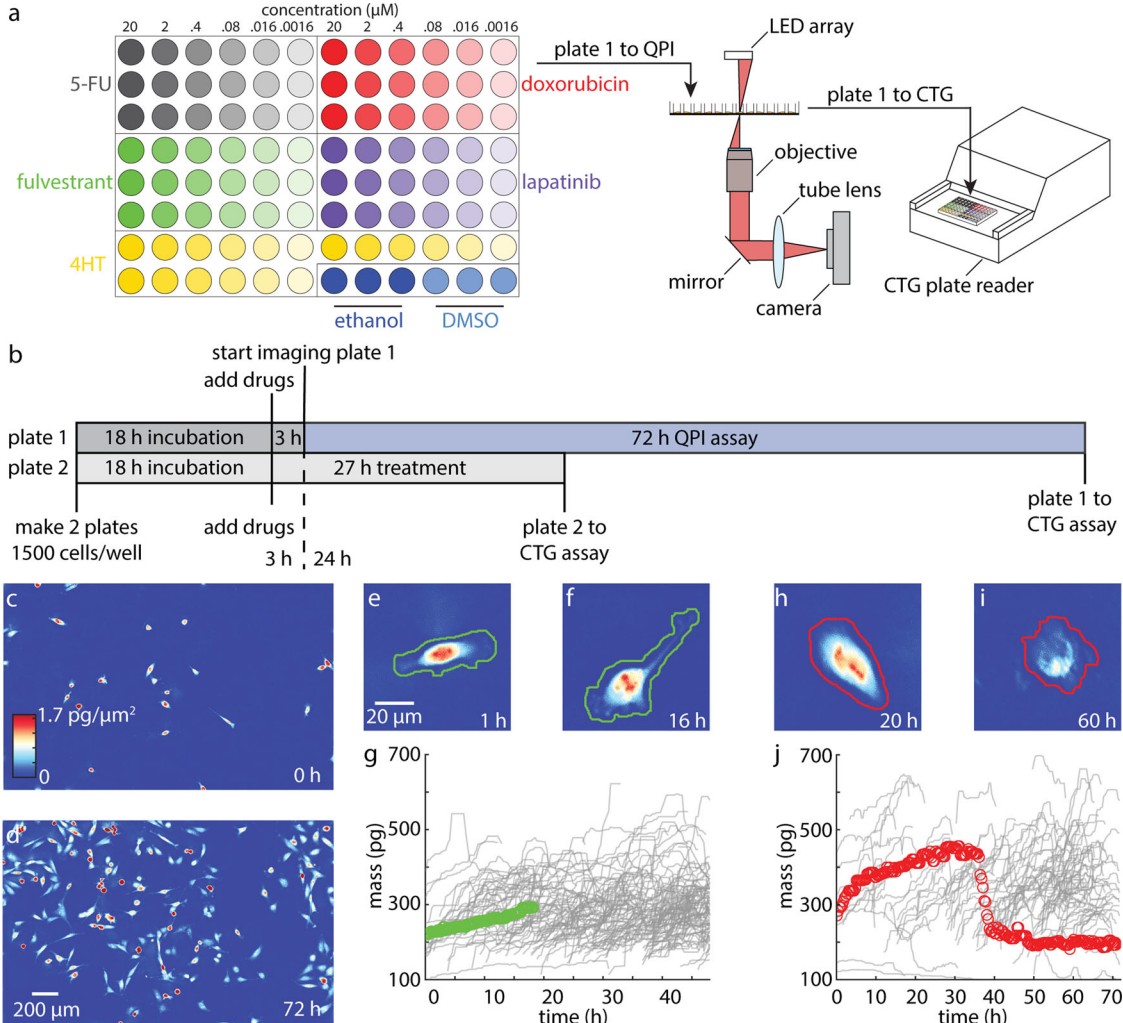

**Fig. 1 QPI measures drug sensitivity and temporal dynamics for more than 20,000 cells in a single experiment. a** Example plate setup for measuring a 6-point dose response of five cancer therapies in triplicate in addition to two solvent controls (left). Growth response of cells are first measured non-invasively using QPI (center) before being assayed using CTG (right). **b** Experimental timeline showing two plates set up in parallel to measure cell viability using CTG at 27 h after drug exposure (3 h of microscope setup plus 24 h of imaging) and 75 h after drug exposure (3 h of microscope setup plus 72 h of imaging) **c** Representative QPI data from a single location within a single DMSO control well at 0 h, **d** and at 72 h with 10–100 MDA-MB-231 cells imaged per location. Scalebar is 200 μm. **e-f** QPI image of a growing cell in DMSO (solvent control) at 1 h and at 16 h. Automatically segmented boundary for healthy growing cell is shown in green. **g** Corresponding measurement of growing cell mass versus time (green) and other cells from the same well (gray). **h-i** QPI image of a dying cell in 2 μM doxorubicin shown at 20 h while still growing and at 60 h, after the cell death event has occurred. Automatically segmented boundary for dying cell shown in red. **j** Corresponding mass versus time for dying cell (red) and other cells from the same well (gray).

optimized to conduct assays in a 96-well plate with 6-point dose response curves and up to five therapies in triplicate including solvent controls (Fig. 1a, Fig. S1c, Table S3). Two plates were set-up simultaneously for each experiment: one plate was incubated for 27 h post drug exposure prior to CTG analysis (3 h equilibration + 24 h), which measures cell ATP content as a surrogate for cell viability, and the other was imaged for 72 h after 3 h equilibration and followed by CTG (Fig. 1b). For QPI, nine imaging locations were chosen in each well with a minimum of 10 cells or small cell clusters per location and imaged for 72 h (Fig. 1c, d, Fig. S3a–d). Therefore, across all 864 imaging locations, we measured the growth response of 20,000 to 130,000 cells or cell clusters per 72 h experiment (Movies S1–S3). Cells/clusters were automatically segmented ("methods") from background for individual quantification of cell mass differentiating healthy growing cells (Fig. 1e–g, Fig. S3e–j) from arrested or dying cells (Fig. 1h–j, Fig. S3k–p).

QPI mass versus time data have several key features that underlie the ability of QPI to distinguish multiple dynamic characteristics of cell responses to drugs. First, the rate of mass accumulation, or cell growth rate (dm/dt), can be used to characterize cell growth[20,21,31,32,37,38]. In healthy cells, the growth rate is constant as cells accumulate mass during each cell cycle (DMSO control, green line in Fig. 1g, Fig. S3g, j, Movie S4–6). The cell growth rate is typically proportional to the mass of the cell or cluster[25]. We normalized the slope of a linear regression for individual cell/cluster mass versus time by the cell/cluster initial mass to find the specific growth rate (SGR). The SGR, therefore, accounts for variations in growth rate due to differences in cell or cluster size (Fig. S4a–c). For proliferating cells, the resulting SGR matches the exponential growth constant measured by cell counting (Fig. S4d–f) as cells double their mass with each cell cycle[39]. However, when exposed to therapies, mass versus time tracks of individual cells or clusters

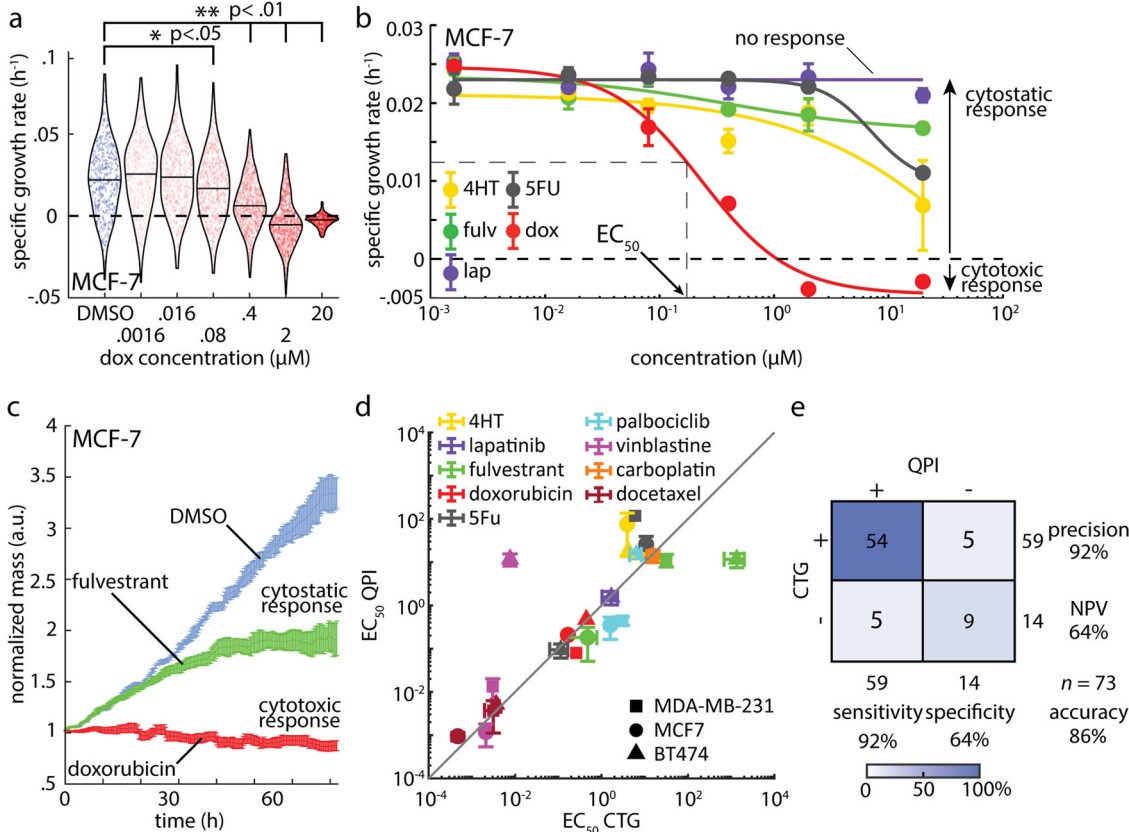

**Fig. 2 Dose response measurement with QPI is concordant to CTG and also characterizes mechanism-dependent depth of response. a** QPI measurements of the specific growth rate distributions for MCF-7 cells at increasing concentrations of doxorubicin. Individual specific growth rate measurements are indicated by points in the distribution. Blue: DMSO control, red: doxorubicin. Black outline is a kernel density function fit to the distribution of specific growth rates for each condition. *$p < 0.05$, **$p < 0.01$. **b** Each point on the dose response curve represents the specific growth rate of 450–700 MCF-7 cell clusters averaged over 27 individual imaging locations used to fit a 4-point Hill equation to dose response data for measurement of $EC_{50}$ and depth of response. Color indicates drug. Cytostatic response is noted as a moderate decrease in SGR, while cytostatic responses are indicated by a minimum asymptote of the hill curve at or below zero. **c** Average mass normalized by the initial mass of each cluster versus time for 486 clusters in response to 20 μM doxorubicin treatment, 615 clusters in response to 20 μM fulvestrant, and 476 clusters treated with DMSO control. Cytostatic response can be observed as the population slows mass accumulation (fulvestrant) while a cytotoxic response results in a gradual loss of mass due to cell death (doxorubicin). **d** Comparison of $EC_{50}$ from Hill equation fitting to CTG and QPI data. Gray line shows the expected relationship ($EC_{50,CTG} = EC_{50,QPI}$). Color indicates drug, symbol indicates cell line. Correlation coefficient, $R = 0.83$, $p = 7.5 \times 10^{-7}$, Concordance coefficient = 0.84. **e** Confusion matrix showing the precision and accuracy of QPI relative to CTG by comparing the frequency QPI predicts the same outcome as CTG. Error bars show standard error of the mean (SEM).

reveal complex dynamic responses. For example, one individual MDA-MB-231 cell exposed to 2 μM doxorubicin exhibits robust growth ($SGR = 0.0358\,h^{-1}$, $t_D = 19\,h$) for the first 10 h before showing reduced growth ($SGR = 0.0105\,h^{-1}$, $t_D = 66\,h$) during the next 26 h, followed by an abrupt decrease in mass during cell death ($SGR = -0.217\,h^{-1}$) and a subsequent gradual loss of mass ($SGR = -0.0033\,h^{-1}$, Fig. 1h–j, Fig. S5, Movie S7). These dynamics are highly heterogeneous from cell to cell or cluster to cluster (Fig. 1f, Fig. S3h,p red versus gray traces, Movie S8–10). A broad range of dynamic behaviors is thus captured by label-free, single-cell QPI data during drug response measurements.

**Determination of sensitivity, $EC_{50}$, and depth of response (DoR).** A dose-dependent change in growth, as indicated by a change in the rate of mass accumulation or loss, is the first key parameter that can be extracted from QPI dose-response data[21,32]. For example, the SGR distribution for MCF-7 cells decreases with increasing doxorubicin concentration to a minimum of $-0.003 \pm 0.004\,h^{-1}$ (mean ± standard deviation, SD) at 20 μM, the highest dose tested, indicating cellular response to the drug (Fig. 2a). We fit the average SGR of each condition to a

sigmoid dose-response curve using the Hill equation[40,41]. The computed logistic fit parameters using the 4-parameter Hill model for all conditions are available as Supplementary Data 1. To determine if there is a response, we test for goodness of fit adjusted for degrees of freedom using an F-test at significance $p < 0.01$ by comparison to a flat line, as an indicator of no response (Fig. 2b). For cases with a better fit to the Hill equation, the resulting 4-parameter Hill curve can be used to determine the $EC_{50}$, a measure of sensitivity to therapy because it represents the effective concentration at which 50% of the cells respond to therapy (Fig. 2b, Fig. S6). The $EC_{50}$ for a particular therapy is the inflection point of the Hill curve, and one of the fitting parameters in the Hill equation (see Methods).

For conditions with a response, the DoR is computed from the fitted Hill curve as the difference between the asymptotes at the highest and lowest concentrations, normalized by the asymptote at low concentration. This normalization accounts for differences in the control growth rates of each cell line. The DoR is used to determine how toxic a therapy is to a particular cell line (Fig. 2b). A DoR less than 1 represents a cytostatic response with reduced cell growth relative to control; a DoR greater than 1 represents a

cytotoxic response resulting in loss of mass and is associated with cell death (Fig. 2c). The $EC_{50}$ measured using QPI is highly concordant to the $EC_{50}$ measured using 72 h CTG as a gold-standard of drug response[14,42] with a correlation coefficient of 0.83 ($p < 0.001$) (Fig. 2d) and concordance coefficient of 0.84 (95% confidence interval = [0.57, 0.98]). Additionally, predictions of cell line/compound pairs that show no response ($n = 14$) versus those showing a response ($n = 59$) are highly concordant with CTG 72 h results (86%, Fig. 2e). When comparing QPI to CTG, concordance at 24 h and 72 h time points was generally high for $EC_{50}$ values but low for DoR, indicating that drugs can affect cell growth distinctly from measurable changes in ATP (Fig. S7).

**Measurement of time of response (ToR) from dynamic QPI data**. By precisely measuring the mass of individual cells during treatment, QPI gives rapid and sensitive insight into the dynamic response of cells to therapy throughout the experiment with high temporal resolution. For example, the normalized mass versus time for BT-474 cells treated with 20 μM of palbociclib, 0.016 μM of docetaxel, and 20 μM of vinblastine, the nearest concentration tested above the measured $EC_{50}$s, initially behave similarly to the control, but then exhibit differential, time-dependent responses (Fig. 3a). Of these conditions, palbociclib, a CDK 4/6 inhibitor that prevents the transition from G1 to S phase, elicits the fastest response, within 5 h from the start of imaging (8 h post-exposure), and with the largest DoR as indicated by a reduction in mass (Fig. 3a). The temporal dynamics of response are also dose-dependent, with higher concentrations resulting in a substantially faster response than lower concentrations as larger proportion cells respond more quickly at higher doses (Fig. 3b, Fig. S8a, b). To study response dynamics, we examined the changing distributions of SGR of single cells or cell clusters as a function of time (Fig. 3c, Fig. S8c, d). In the case of BT-474 cells treated with 20 μM vinblastine, for example, we observed that the average growth rate slowly decreases over 12–60 h of treatment, as indicated by a gradual separation of the control and treated cell populations (Fig. 3c). We then used the Hellinger distance to quantify the time required for cells to respond to therapy (Fig. 3d). Hellinger distance provides a measure of the dynamic response of cells relative to any time-dependent changes in plate-matched solvent controls and is impacted by changes in both the mean response as well as the variance of the distributions[43] (Fig. S9). This enables us to quantify the ToR consistently across conditions and cell types. We defined the ToR as the time point when Hellinger distance crossed the threshold set by the maximum Hellinger distance of the solvent control from the untreated control (Fig. 3d, Fig. S10a–c). Often, the concentration tested just below the $EC_{50}$ elicited no response. As expected, drugs that elicited no response never crossed our Hellinger distance response threshold (Fig. S10d–f). We, therefore, used the ToR at the tested concentration just above the calculated $EC_{50}$ as the nearest approximation of ToR at the calculated $EC_{50}$ (ToR at $EC_{50}$). Comparing ToR to DoR indicates that cytotoxic conditions elicited the fastest response, but even conditions classified as cytostatic often elicited a response in less than 24 h with a moderately negative relationship between ToR and DoR ($R = -0.62$, $p = 0.002$, Fig. 3e). Plotting the ToR against the $EC_{50}$ further classifies drug-cell pairs based on sensitivity and the speed of response (Fig. 3f).

**Measurement of heterogeneity and tracking of outliers**. SGR in control populations shows intrinsic heterogeneity as shown by the large standard deviation in growth rates even in the control group (Fig. S11a–c). This intrinsic heterogeneity is impacted by

treatment, with some drugs reducing heterogeneity by as much as fourfold, as in MDA-MB-231 with 20 μM doxorubicin (Fig. 4a, Fig. S11d, e). Additionally, cell-to-cell heterogeneity evolves over time during drug treatment, and this change is captured by QPI. For example, MDA-MB-231 cells treated with 20 μM docetaxel show a gradual change in SGR distribution over 72 h with a distinctive long tail of non-responders that persists at 72 h, despite treatment with a cytostatic compound for >2.5 cell cycles (Fig. 4b). Since QPI is based on longitudinal tracking of cells from microscopy images, we can track individual non-responders backwards through the duration of the experiment (Fig. 4b, c). For example, select MDA-MB-231 cells treated with 20 μM docetaxel show growth that is distinctly different from the periodic doubling of cell mass observed in control populations (Fig. 4c). This results in cells with a distinctively large mass as compared to control cells (Fig. 4d–f, Movie S10). Given that docetaxel primarily acts as a microtubule inhibitor, QPI data imply that treated cells, unable to divide or undergo apoptosis, reenter the cell cycle and continue accumulating mass at a similar rate as the control (Fig. 4c). This change in heterogeneity is dose-dependent (Fig. 2a), with an increase in dose corresponding to a reduction in cell-to-cell heterogeneity of growth (Fig. S12). Such impacts are drug-specific. Some compounds induce a significant decrease in the mean response, but no significant change to the spread (i.e., SD) within the population, even at high concentrations of therapy, such as in BT-474 treated with 20 μM docetaxel and 20 μM vinblastine (Fig. S11d). We used the SD at the tested concentration nearest the $EC_{50}$ (SD at $EC_{50}$) as a relevant measure of heterogeneity in a responding cell population. There was little relationship between the measured heterogeneity during treatment (SD at $EC_{50}$) and $EC_{50}$ or ToR, indicating that the impact of drugs on growth heterogeneity provides a measurement of drug response that is independent of sensitivity and speed of response (Fig. 4g, Fig. S13). We quantified the limit of QPI to identify the proportion of resistant cells in an in silico mixture[33], using precision-recall analysis[44–46] (Fig. S14a). Precision-recall curves have previously been shown to be an appropriate measure that is superior to receiver operating characteristics for evaluating a classifier for unbalanced classes[44–46]. This is especially important in this case where we are considering a fraction of resistant cells that are small relative to the size of the population. We found that QPI can identify resistant cells in proportions as low as 0.1–2%, however, this is highly dependent on initial SGR, DoR, cell line, therapy, and concentration (Fig. 4h, Fig. S14b).

**Multiparametric measurements of cancer cell drug response with QPI**. To further elucidate the relationships among QPI parameters, we computed the Pearson correlation coefficient between all QPI measured parameters ($EC_{50}$, DoR, ToR at $EC_{50}$, and SD at $EC_{50}$), as well as CTG-based $EC_{50}$ and DoR (Fig. 5a, Fig. S15). We found that the ToR at $EC_{50}$ has a moderate negative correlation to DoR ($R = -0.62$, $p = 0.002$) indicating that more toxic drugs tend to cause a decrease in the time it takes to elicit a response (Fig. 3e, Fig. 5a). We determined the $EC_{50}$ for heterogeneity based on 4-parameter Hill equation fitting (Fig. S12) and found it was strongly correlated to the $EC_{50}$ measured for cell growth ($R = 0.76$, $p = 0.01$), (Fig. 5b). We also found the SD at 20 μM is strongly correlated to the SD at $EC_{50}$ indicating that increased concentration beyond the $EC_{50}$ does not cause heterogeneity to decline beyond its level at the responding concentration ($R = 0.76$, $p < 0.001$, Fig. 5c). However, we found low correlations between the change in heterogeneity, the ToR, and the $EC_{50}$. Furthermore, dimensional reduction using principal

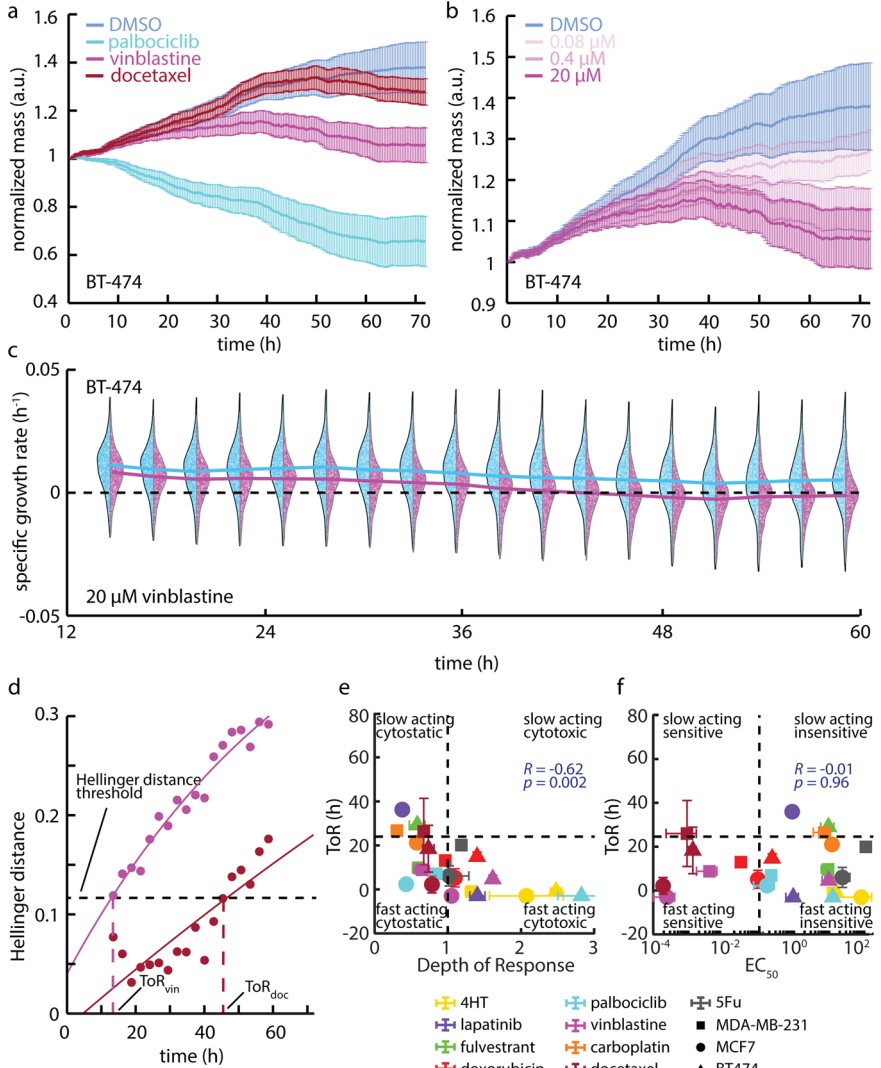

**Fig. 3 QPI reveals temporal dynamics of response to therapy. a** Mass versus time normalized by the initial mass averaged over all BT-474 clusters at each time point for 20 μM palbociclib, 20 μM vinblastine, and 0.016 μM docetaxel (nearest concentration to the measured $EC_{50}$). Error bars show SEM. **b** Mass versus time normalized by the initial mass averaged over all cells at each time point for 0.08 μM, 0.4 μM, and 20 μM of vinblastine. Error bars show SEM. **c** BT-474 response to 20 μM vinblastine (magenta, $n = 1944$ cells) versus DMSO control (blue, $n = 2414$) in 24 h bins centered on different time points show an initially similar distribution that slowly begins to deviate over time. Solid lines represent the median of the distribution as a function of time. Individual data points show the specific growth rate of individual cells within population distributions. **d** Hellinger distance, a measure of the similarity between two probability distributions, versus time for 20 μM vinblastine and 0.016 μM docetaxel quantifies the difference between each drug-treated group and the control to identify when the difference is significant enough to determine the ToR as shown by the threshold. Black dashed line represents the threshold determined by the maximum Hellinger distance between the DMSO and untreated control, the magenta dashed line shows the ToR for vinblastine, the maroon dashed line shows the ToR for docetaxel. **e** ToR near $EC_{50}$ versus depth of response (DoR) from QPI data classifies each drug based on its cytotoxicity and how quickly it affects cell growth. Vertical dashed line is at DoR= 1 as the threshold between a cytostatic and cytotoxic response. Horizontal dashed line is at ToR = 24 h, as the division between fast and slow-acting drugs. **f** ToR near $EC_{50}$ plotted against the $EC_{50}$ classifies responses as fast or slow relative to drug sensitivity. The shape of each data point shows the cell line and the color describes the drug condition with error bars showing SEM in **e** and **f**.

component analysis (PCA) suggests that $EC_{50}$ and DoR alone only account for about 70% of the information present in the data with the other 30% being split between SD and ToR. Thus, this group of four parameters derived from QPI provides mostly orthogonal measurements that independently describe different aspects of how cells respond to therapy (Fig. S15).

To study how growth rate and heterogeneity change as a function of time, we parameterized these two variables by time and observed unique drug- and dose-dependent behaviors (Fig. 5d, e). For example, we found that the change in heterogeneity for both

MDA-MB-231 and BT-474 cells occurred simultaneously with a change in growth rate, but when treated with 4HT, an estrogen receptor-targeted therapy that should affect ER-positive lines such as MCF-7, their behaviors were quite different (Fig. 5d, e, Fig. S16). MDA-MB-231 cells treated with 4HT first experienced a reduction in growth rate before heterogeneity was affected (Fig. 5d). Once the cells started dying, the heterogeneity of the population decreased dramatically at a constant growth rate. BT-474 cells however increased in heterogeneity while their growth rate was decreasing (Fig. 5e). We found that more generally, MDA-MB-231 cells

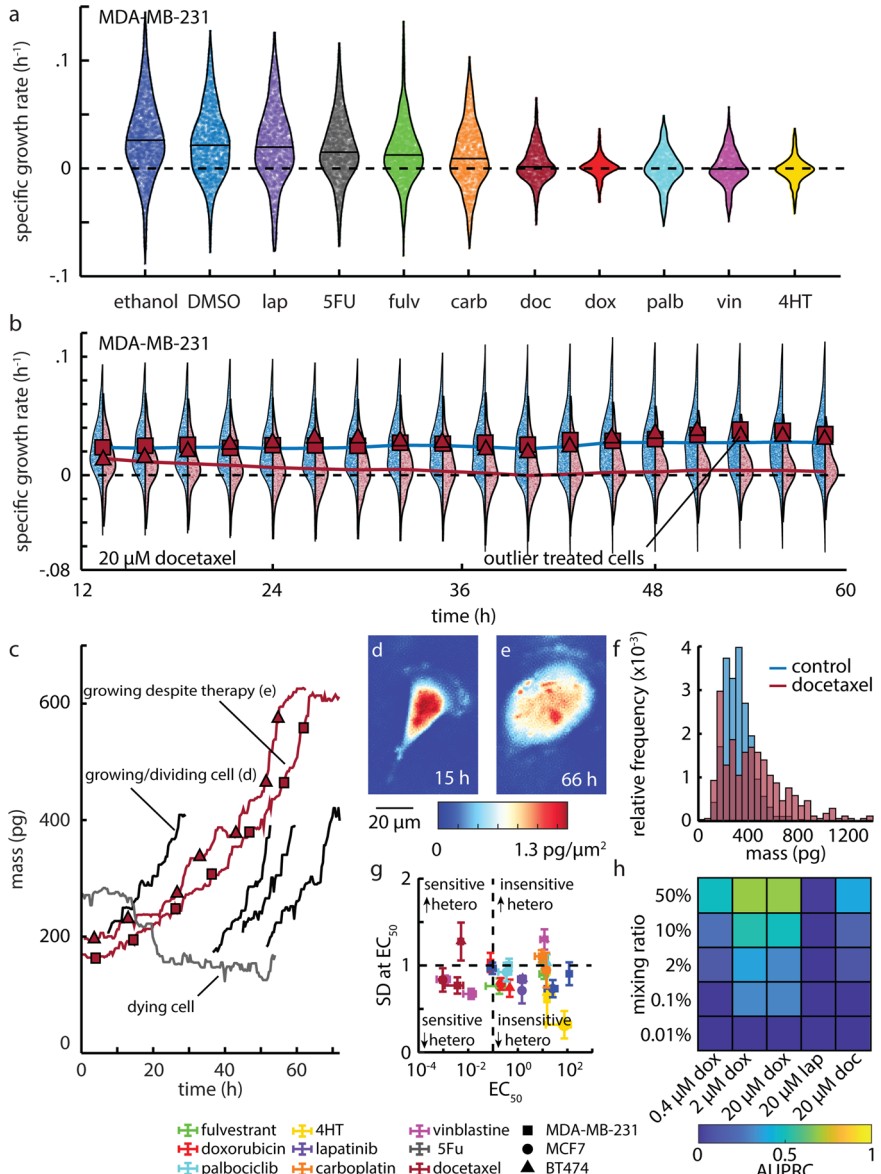

**Fig. 4 QPI measures growth heterogeneity in healthy and drug-treated populations. a** Growth rate distribution of 21,791 MDA-MB-231 cells at the end of 72 h of 20 μM of the indicated drug exposure. Individual cell data are plotted and bound by the kernel density function of the distribution (black outline). Mean population response is shown as a horizontal line. Dashed line shows a growth rate of zero. **b** MDA-MB-231 population growth rate distributions throughout during 72 h treatment with DMSO (blue, n = 1954 cells) and 20 μM docetaxel (maroon, n = 1071 cells). Two cells, with growth rates greater than mean of control (growth rate shown as square and triangle) at the end of experiment are traced back in time to determine how their growth rate evolved throughout the experiment. Dashed line shows a growth rate of zero. **c** Mass versus time tracks for these two indicated cells demonstrates that these cells grew robustly throughout the experiment despite the high concentration of docetaxel. A dying cell from this experiment (gray) and normally growing cells from DMSO control (black) are also shown. **d** Control cell from panel c indicating normal cell size and appearance. **e** Aberrantly large cell from panel c persisting in the presence of 20 μM docetaxel. **f** Histogram showing the final mass for docetaxel treated cells (red, n = 343 cells) as well as for DMSO treated cells (blue, n = 563). **g** Standard deviation at EC$_{50}$ plotted against EC$_{50}$ for cell growth. Points below horizontal dashed line (24 h) represent fast responders, vertical dashed line divides sensitivity from insensitivity of a cell line to a particular therapy. Error bars show SEM. Heterogeneity is abbreviated as hetero. **h** Normalized area under precision-recall curve (AUPRC) plotted against the % control cells mixed into the drug-treated population (0.4–20 μM doxorubicin, 20 μM lapatinib, 20 μM docetaxel).

responded to treatment with both reduced growth rate and reduced heterogeneity (Figs. S17, S18a, b), but BT-474 cells responded to treatment with both reduced growth rate and slightly increased heterogeneity for vinblastine, docetaxel, and 4HT (Figs. S18c, d, S19).

## Discussion

We demonstrated the application of QPI as a multiparametric, label-free, high-throughput tool for measuring the growth response of adherent cells to cancer therapies. QPI predictions of which drugs a given population of cells will not respond to and the concentration at which cells demonstrate sensitivity to therapy (EC$_{50}$) is strongly concordant with traditional CTG measurements. Additionally, QPI offers several additional metrics for characterization of drug response at the single cell level. The DoR measured using QPI is a useful tool for classifying the effect of therapies as either cytostatic or cytotoxic. As a method capable of

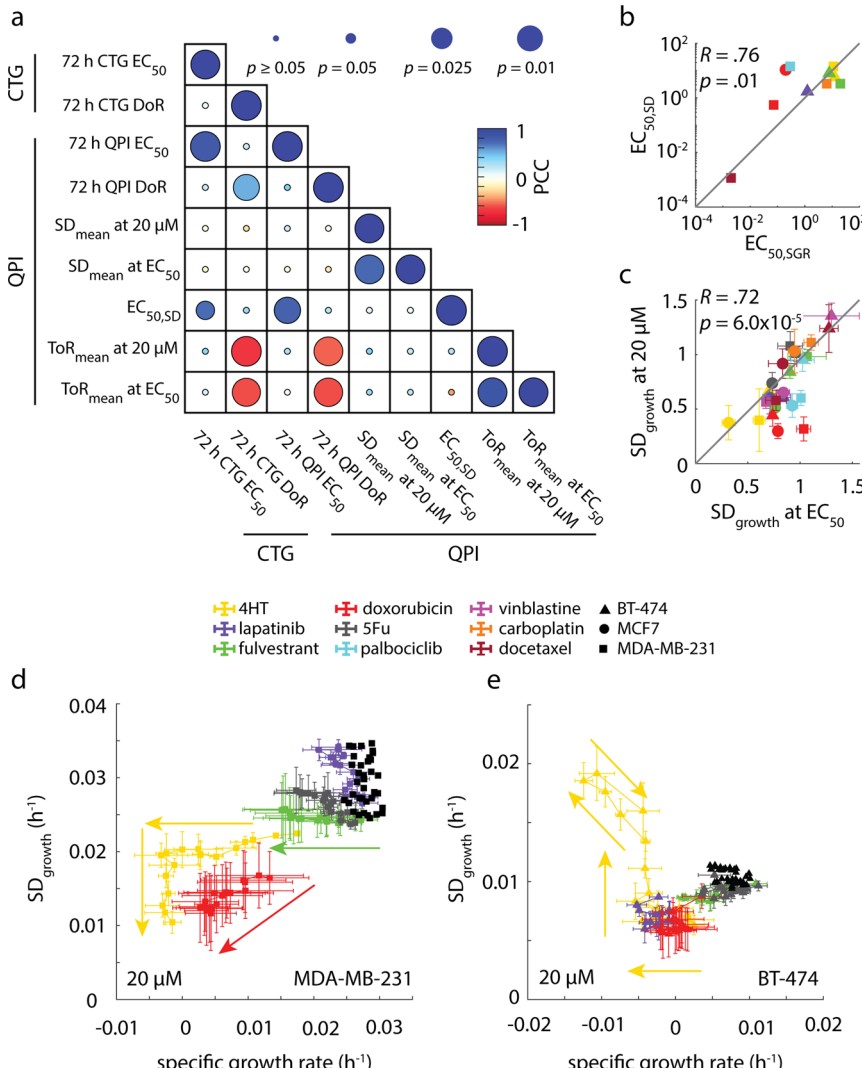

**Fig. 5 QPI functionally classifies drugs based on how they affect cell behavior. a** Correlation matrix showing how functional measurements are related and predictive of how cancer therapies affect heterogeneity. **b** $EC_{50,SD}$ is strongly correlated with $EC_{50,SGR}$ ($R = 0.76$, $p = 0.01$) **c** Standard deviation in growth at 20 μM concentration is strongly correlated with standard deviation of growth at $EC_{50}$. **d** Plot of mean specific growth rate plotted against standard deviation in growth parameterized by time for MDA-MB-231 cells treated with maximal concentration of drug panel A. **e** Plot of mean specific growth rate plotted against standard deviation in growth parameterized by time for BT-474 cells treated with maximal concentration of drug panel A. Control cells for **d** and **e** are shown as a cluster of black dots. Arrows in **d** and **e** show forward direction in time. Error bars show SEM.

tracking growth rates over time, QPI measures response dynamics of single cells, including ToR and heterogeneity, and allows tracking of outliers. Considering all these parameters ($EC_{50}$, DoR, ToR at $EC_{50}$, and SD at $EC_{50}$) reveals significant orthogonality and reveals the dynamic responses of populations over time. A summary of these measured parameters for all studied conditions is available as Supplementary Data 2.

QPI measures the time-averaged growth rate for individual cells/clusters, both on the scale of the entire experiment as well as the growth rate over smaller time intervals demonstrating how QPI tracks the temporal dynamics of growth and heterogeneity (Figs. 3c, 5d, e). This allows QPI to trace those cells back through the assay to determine whether those cells were intrinsically resistant or recovered from the initial growth inhibition by adapting to its presence in the environment (Fig. 4b). We also found both MCF-7 cells and BT-474 cells to be sensitive to lapatinib, a therapy targeting the HER2 receptor, with an $EC_{50}$ of ~1.5 μM. Although the sensitivity to lapatinib for the two cell lines was very similar, BT-474, a HER2+ cell line, shows a

cytotoxic response, a decrease in heterogeneity by 30% relative to MCF-7, and a ToR that is 60 h faster than the cytostatic response of MCF-7. We also found that there was little ability for QPI to distinguish resistant cells in the treated MCF-7 population, however, QPI was able to identify control cells mixed into the treated BT-474 population down to a mixing ratio of 0.1% (Fig. S14b). Together, this demonstrates how these multiple parameters together show how much more effective lapatinib is against BT-474 than MCF-7 cells than sensitivity alone. This level of insight is crucial for understanding how resistant cells and subpopulations develop and for identifying therapies that overcome resistance.

There are several key aspects of multiparametric QPI that point towards clinical applications, in addition to its concordance with CTG, which is already widely used in clinical trials of functional oncology[10,11,14]. QPI is a direct readout of cell response and is marker free. It is therefore not susceptible to drug-stain interference or false positives from sub-lethal/sub-cytostatic alterations in ATP production. QPI requires relatively few cells, making

it amenable for application to clinical samples with limited cell numbers or where expansion is difficult. The experiments described here required ~200,000 cells to achieve sufficient density for imaging in each of 96-wells, with QPI measuring the response of up to half of the total cells plated. Overall, even though previous work has studied heterogeneity by mixing resistant with sensitive cells[33], we believe an important direction for future work is to investigate the parameters presented here in the context of mixed populations.

We also showed that 72 h QPI experiments are not always required for measuring cell sensitivity to therapy. In 22/24 cases, 24 h QPI was able to predict treatment response relative to CTG and 24 h QPI results are strongly correlated with 72 h QPI results (Fig. S7). This is consistent with ToR data showing that 16 conditions elicited a response near the $EC_{50}$ concentration in less than 24 h. Taken together, this suggests that QPI can be applied to classify drug responses as either fast or slow responders based on ToR and then these data can be used to streamline testing of clinical samples. As part of a clinical workflow, this would require plating samples on two or more plates and starting all drug treatments at the same time after plating. Fast responders would be imaged for the first interval (ex. 24 h) and slow responders for the next period (ex. 24–48 h). In this way, QPI can be applied to rapidly quantify drug responses to a larger panel of drugs with varying mechanisms in as little as 48 h. Overall, our work here indicates that the rich, quantitative data on cell responses measured by QPI can generate new insights into drug response that may ultimately inform clinical decision-making.

## Methods

**QPI.** QPI was performed with a custom-built differential phase contrast (DPC)[35,36] microscope. The microscope was built according to the drawing in Fig. S1a and the necessary components are listed with part numbers and suppliers in Table S2 and placed inside a cell culture incubator for temperature, humidity, and 5% $CO_2$ control. Images were acquired using a 10×, 0.25 numerical aperture ($NA_{obj}$) objective and Grasshopper3 USB camera containing 1920 × 1200 pixels that are 0.54 μM in size (Teledyne FLIR, Wilsonville, OR). Exposure time was set to 50 μs and the gain was 25 dB. Focus was maintained with an automated focusing algorithm[47]. We used a high-speed $xy$ translation stage (MLS203, Thorlabs, U.S.A), set to an acceleration of 2480 mm/s[2] and a maximum velocity of 400 mm/s, to image each location with a temporal resolution of 20 min. For illumination, we used an 18 mm square 8 × 8 light emitting diode (LED) array, positioned 24 mm above the sample plane allowing for a numerical aperture of illumination ($NA_{LED\ array}$) of 0.39. The coherence parameter, denoted as $\sigma$, which is the ratio of $NA_{LED\ array}$:$NA_{obj}$ was 1.52[35,36]. The LED array was controlled via an Arduino Metro M4 (Adafruit, U.S.A.) We implemented DPC image acquisition and phase retrieval as described by Waller et al.[35,36]. We captured four images with half circle illumination (top, bottom, left, right) in less than one second including microscope motion. Opposing pairs of images were used to compute the phase gradient in two orthogonal directions. We then computed the phase shift by deconvolution with the estimated optical transfer function via Tikhonov regularization[30,31]. Key parameters for obtaining the phase reconstruction include the illumination angles (90 and 180 since our images were taken along axes normal to each other) and the regularization parameter ($1 \times 10^{-3}$), which was determined experimentally based on system calibration (described below).

**Controlling for solvent toxicity and phototoxicity.** We controlled for the effect of drug solvents and phototoxicity using on-plate solvent controls during each experiment[48]. These were matched to the highest concentrations of solvent used in the experiment (0.125%). Both plates had DMSO controls (Fig. 1a, Fig. S1c) as most (8 out of 9) compounds tested were solubilized in DMSO. Ethanol control wells were used on plate 1 (Fig. 1a) as 4-hydroxy-tamoxifen was solubilized in ethanol. Additionally, we found the power of the LED array at the sample plane to be ~790 nW integrated over a single field of view corresponding to a flux of $4 \times 10^6$ photons per μm[2]. This is far lower than the $5 \times 10^8$ photons per μm[2] previously reported to be considered a safe exposure[49], and at a longer wavelength, 624 nm here vs. 473 nm, and consequently lower energy per photon. Finally, comparison among solvent controls, untreated controls, and cell counting performed on replicate plates (under no illumination) yielded no significant difference in control growth rate (Fig. 4f).

**Calibration of phase measurements.** We calibrated the DPC QPI microscope prior to imaging, to validate the system and to ensure the measurements were not impacted by misalignment[50]. The first calibration experiment was to validate the

centering of the LED array by comparing the raw intensity from each half of the LED array while imaging an empty sample. After centering and aligning the LED array we found that the intensity from each half of the LED array was equal to 5% of the mean for all four half circles (Fig. S2a). We calibrated the phase measurements by imaging polystyrene microbeads (Polysciences, Warrington, PA), a common calibration standard for QPI microscopes[51–54]. To prepare our bead sample, we diluted the bead stock by a factor of 10 such that our bead solution was ~0.25% microbeads (weight/volume). We mixed the solution using a vortex for 10 s and then pipetted 100 μL onto a glass slide. After the water evaporated, we pipetted 50 μL of NOA73 (Norland Optical Adhesives, $n = 1.56$) on top of the beads ($n = 1.583$) and flattened the polymer with a coverslip to obtain an even layer of polymer. We cured for 1 min in a UV oven to obtain a sample with a difference in refractive index comparable to that between cells and cell culture media ($\Delta n = 0.023$). We imaged the beads using QPI (Fig. S2b) and found that mean refractive index of the beads was ~1.584, within 0.1% of the previously reported value of 1.583 for 624 nm light[55] (Fig. S2c). Beads were imaged 100 times in a 3-min interval to measure the temporal precision, and we found temporal coefficient of variation was 5.4% (Fig S2c).

**Image processing.** Cells were segmented using a Sobel filter to find cell edges, and morphological operators to create a mask. Single MDA-MB-231 cells were further segmented using a watershed algorithm. Cells were masked and an 8th order polynomial fit was removed from the background prior to averaging images from each experiment to correct for aberrations and optical artifacts. A rolling ball filter, using a disk structuring element of 100 px, was applied to remove high spatial frequency noise. Cell mass was then computed using a cell average specific refractive increment of $1.8 \times 10^{-4}$ m[3]/kg[23]. Segmented cells were tracked from frame to frame based on approximate minimization of the distance between cell objects in successive frames in terms of cell mass and position in $x$ and $y$[56].

**Drug mixing and dilution.** Prior to the start of the experiment, drugs were mixed with the appropriate volume of solvent to make a 20 mM stock solution. Stock solutions were stored in a −20 °C freezer and were never thawed more than five times to preserve the efficacy of the therapies. The stock concentration of the therapies was then aliquoted into media to make a 40 μM solution, which was serially diluted on 96-well plate with 1 mL wells to make a solution that is double the desired concentration for the assay. The diluted therapies were added to cells 3 h prior to the start of the assay at a 1:1 ratio of drugged media to cell media, to dilute the concentration of therapy to its final concentration.

**QPI assay.** 1500 cells were plated in each well of two 96 well plates with 100 μL of media to allow space for drugged media to be added. Cells were incubated in cell culture conditions for 18 h prior to dosing. 100 μL of diluted cancer treatments and solvent controls were added to the cells 3 h prior to the start of imaging. Cells were allowed to incubate on the microscope in cell culture conditions (37 °C and 5% $CO_2$) for an hour prior to focusing the center of each well. Nine imaging positions were selected per well and each location was imaged every 20 min with a single autofocus before each imaging cycle to account for thermal and z-stage drift. After 24 h of imaging plate 1, the first CTG assay was performed on plate 2 (Fig. 1b) while continuing to image plate 1 for a total of 72 h. After 72 h of imaging, the second CTG assay was performed on plate 1.

**Cell counting.** Cell counting experiments were done by measuring proliferation throughout the duration of an experiment. Cells were counted in three different ethanol-treated wells at 0 h, 18 h, 36 h, 48 h, and 72 h to measure the doubling time for MCF-7 and MDA-MB-231 cells. The following equation was used to compute the exponential growth constant for each cell line:

$$SGR = \frac{\ln(2)}{t_{doubling}} \qquad (1)$$

Such that $t_{doubling}$ is the doubling time measured using cell counting for each cell line, ln(2) is the natural logarithm of 2, and SGR is the specific growth rate also known as the exponential growth constant.

**Cell culture.** All cell lines were acquired from ATCC and routinely screened for mycoplasma infection using the Agilent MycoSensor qPCR assay. MCF-7 cells were cultured in Dulbecco's Modified Eagle Medium F12 supplemented with 10% heat-inactivated fetal bovine serum (FBS) and 1% penicillin/streptomycin (Pen–Strep). MDA-MB-231 cells were cultured in RPMI medium supplemented with 10% FBS and 1% Pen–Strep. BT-474 cells were cultured in Hybri-Care Medium 46-X prepared with 18 MΩ deionized water supplemented with 1.3 mM of sodium bicarbonate, 10% heat-inactivated FBS, and 1% Pen––Strep. Cells were passaged on 10 cm cell culture treated dishes at 37 °C/5% $CO_2$ and passaged by washing with Dulbecco's phosphate buffered saline and then incubating with Trypsin at 37 °C with 5% $CO_2$ for 7 min before splitting at a 1:5 ratio.

**CTG assay.** CTG is a cell viability assay that quantifies the amount of ATP present at the end of an experiment as an indicator of the number of live cells. The ATP

content measured in each condition is readout by a luminescent signal, which is normalized by the luminescence of the control in order to determine cell viability relative to on-plate controls. We performed the CTG assay by first removing 100 μL of media from each well on the plate, which was then replaced with an equal amount of CellTiter-Glo reagent (Promega, G7572). Assayed plates were shaken at 500 RPM for 20 min and allowed to rest for 10 min. 100 μL of volume from each well was transferred to a white 96-well plate (Perkin Elmer, 6005680). Luminescence data were then collected from each well using an Envision plate reader (Perkin Elmer) and normalized against the solvent control to measure ATP content. We fit a 4-parameter hill curve to the dose response to compute the $EC_{50}$ and depth of response.

**Statistics and reproducibility**. We measured the correlation between variables using the Pearson correlation coefficient as implemented in Matlab which tests the null hypothesis that there is no relationship between the variables. We also used Lin's concordance coefficient to measure the concordance between variables[57]. We computed the confidence interval by bootstrapping based on resampling the observed data 10,000 times and reporting the confidence interval as the minimum and maximum of the middle 95% of these data.

**In silico mixing analysis**. We simulated the mixture of resistant and sensitive cells by randomly selecting control cells to mix into the drug-treated population at mixing ratios of 50%, 25%, 10%, 5%, 2%, 1%, 0.1%, and 0.01%. We used this mixture to identify resistant cells using threshold growth rates ranging from $-0.1\,h^{-1}$ to $0.15\,h^{-1}$ and for each threshold computed the precision and recall of the identified resistant cells. We, therefore, parameterized the precision and recall measurements by threshold-specific growth rates and used the area under the precision-recall curve (AUPRC)[44–46]. Generally, a higher AUPRC indicates a more effective classifier. To compare AUPRCs across mixing ratios we need to compare them to the no-skill line[58]. Therefore, all AUPRCs reported are normalized by the area above the no skill line, which represents the best possible performance of the classifier at a specified mixing ratio. To reduce errors due to random sampling of the control population, especially at the lowest mixing ratios, we repeated the mixing 100 times for each condition and sorted the data based on the measured AUC. We then plotted the median precision and recall value for each tested threshold and reported the median AUC.

**SGR calculation and filtering tracks for goodness of fit**. Cell mass versus time data with a minimum length of 20 frames were median filtered, with a kernel size of 5 frames, to remove small fluctuations. Data with a mean mass lower than 110 pg were removed from the analysis, as these were found to be debris. We time-shifted each mass over time plot, such that the first mass measurement of each cell starts at $t = 0$. Linear regression was applied to find the slope, which defines the growth rate (pg/h), and the y-intercept, which was used as the initial mass. SGR is then the growth rate divided by the initial mass. The standard error of the estimate ($s_{y.x}$) and normalized slope (specific growth rate; $k$) were used to remove outliers. An outlier was defined as 3 median absolute deviation from the median.

**Normalized mass versus time**. The overall mass of an imaging location is the summation of the mass of individual pixels after background correction. Overall mass is collected for each location over time, median filtered, then normalized by the mass at the state of the imaging. All locations in all wells for a given condition were averaged. The standard deviation was calculated for each triplicate of wells.

**Logistic fitting and depth of response calculation**. Average SGR data for individual treatments were fit to both a response (Hill equation) and a no-response (flat line) model. The response model is a four-parameter logistic (Hill equation) function for fitting SGR versus concentration, $C$:

$$\text{SGR} = E_{max} + \frac{E_0 - E_{max}}{\left(1 + \left(C * \left(EC_{50}\right)\right)\right)^{HS}} \tag{2}$$

Such that $E_0$ is asymptote at lowest concentration, $E_{max}$ is asymptote at maximum concentration, $EC_{50}$ is the inflection point of the hill curve, and HS is the Hill slope. The no-response model is a flat line parallel to the concentration axis. The residual variance from each fit was then compared using an F-test with a $p$ value of 0.01. For responding conditions (logistic fit better than no-response fit as determined by F-test), depth of response (DoR) was computed as

$$\text{DoR} = \frac{E_0 - E_{max}}{E_0} \tag{3}$$

**Temporal growth dynamics**. To measure dynamic changes in SGR over time, cell mass versus time tracks were first broken up into overlapping 24 h intervals centered on each imaging time point in the experiment for a total of 144 total intervals. The tracks within each interval were then filtered for a minimum path length of 20 frames within each interval, a minimum mean mass of 110 pg, and goodness of fit to a linear model as described above. The specific growth rate for each cell in each time interval was then found by time shifting each track to start at $t = 0$, and then

using a linear regression to find the rate of mass accumulation and this slope was normalized by the y-intercept of each regression of the time-shifted data in the interval. We binned 8 adjacent intervals together to produce 18 bins throughout the duration of the experiment. We computed the kernel density function in each bin using the mean SGR of each cell in the bin.

**Hellinger distance measurements**. We measured the Hellinger distance between growth rate distributions by fitting a probability density function (PDF) to the distribution of raw growth rates of each cell or cluster normalized by its initial mass such that an integral over the PDF is equal to 1. The Hellinger distance is defined as[43]:

$$H^2(f,g) = 1 - \int \sqrt{f(x)g(x)}dx = \frac{1}{2}\int \left(\sqrt{f(x)} - \sqrt{g(x)}\right)^2 dx \tag{4}$$

which is then discretized to:

$$H^2(S,T) = \frac{1}{2}\sum\left(\sqrt{s_i} - \sqrt{t_i}\right)^2 \tag{5}$$

Such that $H$ is the computed Hellinger distance between probability distribution functions $s$ and $t$. We computed this sum using a histogram bin size of $10^{-4}\,h^{-1}$ so that the bin size was small enough to capture differences in growth rate for BT-474 cells, the slowest growing cell line we tested.

**ToR calculation**. We computed the ToR by plotting the Hellinger distance between the drug-treated group and the control against time, and fitting the data to the following equation:

$$H = a - b * e^{ct} \tag{6}$$

Such that $a$, $b$, and $c$ are fit parameters to minimize the sum of least squared residuals. We fit this model to the Hellinger distance measured between the controls to find the Hellinger distance threshold as the maximum distance between controls for each cell line. We computed the ToR by first fitting this model to the Hellinger distance vs time for each therapy and analytically solving for $t$:

$$t = \frac{1}{c}\log\left(\frac{a - H}{b}\right) \tag{7}$$

Where $H$ is equal to the threshold Hellinger distance, and $a$, $b$, and $c$ are the fitting parameters for the model.

**Reporting summary**. Further information on research design is available in the Nature Research Reporting Summary linked to this article.

## Data availability
Data that support the findings in this manuscript have been deposited in Figshare: https://doi.org/10.6084/m9.figshare.16840435.v1[59].

## Code availability
Code used for processing raw phase images as well as code for computing the parameters described in this manuscript can be found in the following GitHub repository: https://github.com/Zangle-Lab/MultiparametricQPI.

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

## Acknowledgements

This work was supported by the Office of the Assistant Secretary of Defense for Health Affairs through the Breast Cancer Research Program under Award Numbers W81XWH1910065 (T.A.Z.) and W81XWH191066 (P.S.B.) and NIH U01CA217617 (B.E.W.) and Huntsman Cancer Foundation (B.T.S.) and Susan G. Komen (B.T.S.).

## Author contributions

Conceptualization, formal analysis, validation, and writing—review and editing, E.R.P. and T.A.Z.; methodology and investigation, E.R.P., T.E.M., A.B., T.B.; software, data curation, visualization, and writing, E.R.P., T.E.M.; interpretation of data, E.R.P., T.E.M, B.T.S., B.E.W., P.S.B., T.A.Z.; original draft preparation, E.R.P., T.E.M., A.B., S.D.S., E.C-S., T.B., B.T.S., B.E.W., P.S.B., T.A.Z.; supervision, project administration, resources, and funding acquisition, B.T.S., B.E.W., P.S.B., T.A.Z.; All authors have read and agreed to the final version of the manuscript.

## Competing interests

The authors declare no competing interests.
