## [Peer Review File · Communications Biology]

Reviewers' comments:

Reviewer #1 (Remarks to the Author):

Reviewer's report on the manuscript Title: Multiparametric quantitative phase imaging for real-time, single cell, drug screening in 2 breast cancer. Authors: Edward R. Polanco et al.

The manuscript reports about the possibility to quantify mass versus time of cells by Quantitative Phase Imaging (QPI), thus measuring the growth rate of each individual cell.

In their work authors use the breast cancer cell lines MCF-7, BT-474, and 19 MDA-MB-231 in order to verify QPI measurements. A multiparametric approach for determining response to single agent therapies is investigated. Authors claims rapid determination of drug sensitivity, cytotoxicity, heterogeneity, and time of response for up to 100,000 individual cells or small clusters through a single experiment.

Thus they claims that single-cell responses to candidate therapies is possible by the proposed method thanks to possibility to record and evaluate dynamics of single-cell responses to candidate therapies

Although the study can be of interest of the community the manuscript presents some critical problems as reported in the following comments.

The authors should try to afford those issues in order to clarify and improve their report, otherwise the paper is not suitable for publication.

Main issues:

1) the manuscript completely neglects some approaches based on QPI to detect drug sensitivity of cells behavior in lab-on-chip system based on in-flow configurations.

Authors should mention this and comment, in order to furnish to the readers others approaches that have been developed for similar applications. Thus please take into account the following works:

- "Label-Free Assessment of the Drug Resistance of Epithelial Ovarian Cancer Cells in a Microfluidic Holographic Flow Cytometer Boosted through Machine Learning." ACS omega 6.46 (2021): 31046-31057.

- "Sensing morphogenesis of bone cells under microfluidic shear stress by holographic microscopy and automatic aberration compensation with deep learning." Lab on a Chip 21.7 (2021): 1385-1394.

In the Introduction are mentioned some works about toxicity but one important issue is neglected. What about phototoxicity? During QPI measurements are the authors sure that it can be excluded that light irradiation is not affecting the cells' behaviors and related measurements? Authors should comment and give more details on this issue. They can mention the following paper and others on the topics such as: "Investigating fibroblast cells under "safe" and "injurious" blue-light exposure by holographic microscopy." Journal of biophotonics 10.6-7 (2017): 919-927 and "Cellular uptake of mildly oxidized nanographene for drug-delivery applications." ACS Applied Nano Materials 3.1 (2019): 428-439.

- very few details are given about the system used for the QPI measurements. PLEASE add more

details on performances and specs.

- Supplementary Figure S1 is picture BUT no useful details can be observed in the picture. Maybe it can be useful to replicate there the optical configuration, already shown in the main text. PLEASE add again the drawing of the optical setup in this Figure.

- At line 88 in Results : "...custom QPI microscope (Fig. S1a) based on differential phase contrast (DPC) microscopy"

-

- If the system is a DPC microscope the output is a derivative of the QPI, thus in order to retrieve the QPI phase-maps it is necessary to integrate the DPC phase-maps? How authors have integrated? Why they choose to setting a DPC instead of standard interference microscope or Digital Holography or whichever Ptychography system able to furnish directly the QPI instead of its derivative (DPC)

-

- What about possible misalignments of the ptyco-system? (see this paper: "Misalignment-tolerant Fourier Ptychography." IEEE Journal of Selected Topics in Quantum Electronics 27.4 (2020): 1-17.) and comments on how possible misalignment and displacements can affect the measurements.

-

In general, the work is good and results are of high soundness and thus deserves to be published.

The authors have to revise their manuscript addressing the above issues to make it suitable for publication.

Re-review is necessary.

Reviewer #2 (Remarks to the Author):

In this study, Polanco et al. introduced the well-established QPI by their group as a multi-parametric high-throughput tool for assessing the growth response of adherent cells to various cancer therapies. Specifically, QPI was deployed to determine the drug sensitivity of both non-responders and EC50 concentrations. Their study showed that QPI is concordant with traditional CTG measurements and offers additional analytic metrics to characterize drug response at the single-cell level. For instance, DoR can be used to classify the effect of therapies as either cytostatic or cytotoxic. Also, they showed that QPI could be used to track growth rates over time, i.e., response dynamics of single cells, ToR and heterogeneity, and outliers. Altogether, their study indicated that parameters (EC50, DoR, ToR at EC50, and SD at EC50) of QPI are significant orthogonality and can reveal adherent cells' dynamic responses over time. Overall, this study provides a valuable tool to complement the traditional CTG to evaluate the effectiveness of cancer therapies.

Several issues are listed below to be addressed by the authors:

1. The background of CellTiter-Glo should be briefly explained in the manuscript, facilitating the understanding of the broad readers. Also, in the Introduction, the CTG abbreviation should be elaborated, which is missing in the current version.
2. The sentences in some places are challenging to understand. It is recommended to reorganize them to make the descriptive sentences more well-thought-out.
3. This paper involves many meaningful QPI parameters (SGR, EC50, DoR, ToR at EC50, and SD at EC50). It is recommended to list them in a table and give a detailed description to facilitate the reader's understanding and practice.
4. It will be meaningful to select drug-resistant tumor cells individually to assay various parameters of QPI, discuss whether QPI can effectively identify these cells, and discuss the heterogeneity ratio.

We would like to thank both reviewers for their insightful comments which we believe have strengthened our revised manuscript. We have fully responded to all comments in the revised text, as described below. All revisions are described below and highlighted in the revised manuscript using blue text for new text and ~~striketrough~~ for any text that was removed. We have provided two versions of supplementary information: one in the format required for publication and another with changes marked, as in the main text.

Reviewer #1

Comment 1: The manuscript reports about the possibility to quantify mass versus time of cells by Quantitative Phase Imaging (QPI), thus measuring the growth rate of each individual cell. In their work authors use the breast cancer cell lines MCF-7, BT-474, and 19 MDA-MB-231 in order to verify QPI measurements. A multiparametric approach for determining response to single agent therapies is investigated. Authors claims rapid determination of drug sensitivity, cytotoxicity, heterogeneity, and time of response for up to 100,000 individual cells or small clusters though a single experiment.

Thus they claims that single-cell responses to candidate therapies is possible by the proposed method thanks to possibility to record and evaluate dynamics of single-cell responses to candidate therapies

Although the study can be of interest of the community the manuscript presents some critical problems as reported in the following comments.

The authors should try to afford those issues in order to clarify and improve their report, otherwise the paper si not suitable for publication.

Response 1: Thank you for considering our manuscript and for your thoughtful comments. We have considered each one carefully and believe that responding to your comments has greatly improved the quality of our manuscript.

Main issues:

Comment 2: the manuscript completely neglects some approaches based on QPI to dedcted drug sensitiviyy of cells behavior in lab-on-chip system based on in-flow configurarions. Authors should mention this and comment, in order to furnish to the readers others approaches that have been developed for similar applications. Thus please take into account the following works:

- "Label-Free Assessment of the Drug Resistance of Epithelial Ovarian Cancer Cells in a Microfluidic Holographic Flow Cytometer Boosted through Machine Learning." ACS omega 6.46 (2021): 31046-31057.

- "Sensing morphogenesis of bone cells under microfluidic shear stress by holographic microscopy and automatic aberration compensation with deep learning." Lab on a Chip 21.7 (2021): 1385-1394.

Response 2: Thank you for suggesting these articles for a broader background on the uses of QPI for studying how cells respond to perturbations as it is important for readers to have a broad understanding of various QPI modalities and the broader uses/capabilities of QPI reported in the literature. We added discussion of these two references and this method of drug response screening to the 4th paragraph of the **Introduction**.

Comment 3: In the Introduction are mentioned some works about toxicity but one important issue is neglected. What about phototoxicity? During QPI measurements are the authors sure that it can be excluded that light irradiation is not affecting the cells' behaviors and related measurements measurements? Authors should comment and give more details on this issue. They can mention the following paper and others on the topics such as:

- "Investigating fibroblast cells under "safe" and "injurious" blue-light exposure by holographic microscopy." *Journal of biophotonics* 10.6-7 (2017): 919-927 and

- "Cellular uptake of mildly oxidized nanographene for drug-delivery applications." *ACS Applied Nano Materials* 3.1 (2019): 428-439.

Response 3: Thank you for bringing up the importance of controlling for cytotoxicity due to sources other than the drugs being tested. We added a new *Controlling for solvent toxicity and phototoxicity* section in **Methods** to discuss these issues, and used the suggested references in this discussion, reproduced below:

We controlled for the effect of drug solvents and phototoxicity using on-plate solvent controls during each experiment⁴⁸. These were matched to the highest concentrations of solvent used in the experiment (0.125%). Both plates had DMSO controls (**Fig. 1a**, **Fig. S1c**) as most (8 out of 9) compounds testes were solubilized in DMSO. Ethanol control wells were used on plate 1 (**Fig. 1a**) as 4-hydroxytamoxifen was solubilized in ethanol. Additionally, we found the power of the LED array at the sample plane to be approximately 790 nW integrated over a single field of view corresponding to a flux of 4×10^6 photons per μm^2 . This is far lower than the 5×10^8 photons per μm^2 previously reported to be considered a safe exposure⁴⁹, and at a longer wavelength, 660 nm here vs. 473 nm, and consequently lower energy per photon. Finally, comparison among solvent controls, untreated controls, and cell counting performed on replicate plates (under no illumination) yielded no significant difference in control growth rate (**Fig. 4f**)

Comment 4: very few details are given about the system used for the QPI measurements. PLEASE add more details on performances and specs.

Response 4: We have added substantially more details about the system and its performance to the revised text. This includes:

- a) New supplementary Table S2 listing the critical optical components and part numbers along with discussion in the text (paragraph 1 of section "*Measurement of specific growth rate from QPI data*").

Table S2. List of microscope components		
Part	Supplier	Model/part number
Arduino Metro M4	Adafruit	3382
0.8", 8x8 LED array	Adafruit	870

High speed xy stage	Thorlabs	MLS203
10x Olympus PLAN Objective	Thorlabs	RMS10X
z-translation stage	Thorlabs	SM1Z
Flexible drive shaft	McMaster-Carr	3135K15
Sparkfun 2-phase stepper motor	Mouser	474-ROB-10846
25 mm right angle prism	Thorlabs	PS911
30 mm cage cube	Thorlabs	CCM1-4ER
SM1 cage plate adaptor	Thorlabs	LCP6X
SM2 cage plate	Thorlabs	LCP01
Tube lens ($f = 200$ mm)	Thorlabs	ITL200
ITL200 adaptor	Thorlabs	SM2A20
Ø2" lens tube	Thorlabs	SM2L2
Ø1" lens tube	Thorlabs	SM1L2
Grasshopper3 camera	Teledyne-FLIR	GS3-U3-23S6M-C

- b) We added a new panel, **Fig. S1a**, showing a scale diagram of the system with the arrangement of all the necessary components from the new **Table S2**. We reference these new supplements in the first paragraph of the **Results** section titled, "*Measurement of specific growth rate from QPI data*".
- c) We characterized the illumination power of the QPI system (described in more detail in response to comment 3 above), which is described and compared to previous work in the *Controlling for solvent toxicity and phototoxicity* section of **Methods**.
- d) We characterized the accuracy and precision of the QPI system in a new Supplementary Fig. S2 (discussed in **Methods**) by measuring the refractive index of polystyrene beads and comparing to the manufacturer's specifications.
- e) We added more details about the system including exposure time, camera gain, numerical aperture of illumination, coherence parameter, stage acceleration, stage maximum speed, and position of the LED array above the sample to the *QPI* section of **Methods**.
- f) We included more details about the reconstruction method and key parameters such as the regularization parameter and angles of illumination in response to the *QPI* section of **Methods** as well.

Comment 5: Supplementary Figure S1 is picture BUT no useful details can be observed in the picture. Maybe it can be useful to replicate there the optical configuration, already shown in the main text. PLEASE add again the drawn of the optical setup in this Figure.

Response 5: We added a new diagram to **Fig. S1** showing each of the essential components of the microscope. Each component shown in **Fig. S1** is listed with its part number and supplier in the new **Table S2** that was added in response to Comment 4. We reference this new figure panel and table in the first paragraph titled *Measurement of specific growth rate from QPI data* as well as in the *QPI* section of **Methods**.

Comment 6: At line 88 in Results : "...custom QPI microscope (Fig. S1a) based on differential phase contrast (DPC) microscopy"

- If the system is a DPC microscope the output is a derivative of the QPI, thus in order to retrieve the QPI phase-maps it is necessary to integrate the DPC phase-maps? How authors have integrated? Why they choose to setting a DPC instead of standard interference microscope or Digital Holography or whichever Ptychography system able to furnish directly the QPI instead of its derivative (DPC)

Response 6: Thank you for pointing out this possible source for confusion. Here, we capture opposing pairs of images, from which the gradient of phase is computed, then reconstruct the phase as in previous works, (e.g.¹). To resolve this issue and clarify our choice of QPI methods we have made several changes to the manuscript:

- a) To the first paragraph in the **Results** section “*Measurement of specific growth rate from QPI data*” we clarify that we refer to DPC microscopy to mean the combined imaging and reconstruction method, as described by Tian and Waller. We also added a discussion about the choice of QPI method emphasizing that the simple design of the DPC system has several benefits relevant to this work: a flexible design allowing customization for rapid high-throughput measurements of cell growth, compact design to fit within a microscope incubator, and inexpensive components, promoting more widespread use as a clinical screening tool.
- b) In the *QPI* section of **Methods**, we added more information on how we perform the image acquisition and phase retrieval, including key parameters that we used for phase reconstruction such as illumination angles and regularization parameter.

1. Tian, L. & Waller, L. Quantitative differential phase contrast imaging in an LED array microscope. *Optics Express* **23**, 11394-11403, (2015).

Comment 7: What about possible misalignments of the ptyco-system? (sse this paper: "Miscalibration-tolerant Fourier Ptychography." IEEE Journal of Selected Topics in Quantum Electronics 27.4 (2020): 1-17.) and comments on how possible misalignment and displacements can affect the measurements.

Response 7: Thank you for this remark, we agree in the importance of proper alignment in order to reduce/eliminate errors in phase/mass measurements. We added a new figure (**Fig. S2**) showing calibration data, from measurements taken before data was collected for this manuscript that we did not report in the original manuscript. First, we show data where we measured the refractive index of polystyrene microbeads to characterize the accuracy and precision of the microscope (**Fig. S2a-b**). Our measurements were in strong agreement with previous literature⁵⁵ and the reported optical properties of the NOA73 polymer, in which the beads are embedded.

To ensure that the optical system including the LED array was properly aligned prior to the start of imaging, we measured the intensity from each half circle of illumination and adjusted the position of the LED array until the intensity with each half circle illuminated was equal to within 5% of the mean for all 4 half circles. We added a panel to the new supplementary figure (**Fig. S2c**) showing intensity data indicating that our LED array is properly centered.

This supplementary figure is referenced in the first paragraph of **Results** section *Measurement of specific growth rate from QPI data*. We also added a new section in **Methods** *Calibration of phase measurements* describing this information and including the reference⁵⁰ suggested by reviewer to highlight the importance of proper alignment for the retrieval of QPI images.

References:

50. Bianco, V. et al. Miscalibration-Tolerant Fourier Ptychography. *IEEE J. Sel. Top. Quantum Electron.* **27**, 17 (2021).

55. Ma, X.Y. et al. Determination of complex refractive index of polystyrene microspheres from 370 to 1610 nm. *Phys. Med. Biol.* **48**, 4165-4172 (2003).

Comment 8: In general, the work is good and results are of high soundness and thus deserves to be published.

The authors have to revise their manuscript addressing the above issues to make it suitable for publication.

Response 8: Thank you for this note, we appreciate your consideration of our work and believe that the comments you provided have resulted in a substantially improved manuscript.

Reviewer #2

Comment 1: In this study, Polanco et al. introduced the well-established QPI by their group as a multi-parametric high-throughput tool for assessing the growth response of adherent cells to various cancer therapies. Specifically, QPI was deployed to determine the drug sensitivity of both non-responders and EC50 concentrations. Their study showed that QPI is concordant with traditional CTG measurements and offers additional analytic metrics to characterize drug response at the single-cell level. For instance, DoR can be used to classify the effect of therapies as either cytostatic or cytotoxic. Also, they showed that QPI could be used to track growth rates over time, i.e., response dynamics of single cells, ToR and heterogeneity, and outliers. Altogether, their study indicated that parameters (EC50, DoR, ToR at EC50, and SD at EC50) of QPI are significant orthogonality and can reveal adherent cells' dynamic responses over time. Overall, this study provides a valuable tool to complement the traditional CTG to evaluate the effectiveness of cancer therapies.

Several issues are listed below to be addressed by the authors:

Response 1: Thank you for your recommendations on how we can improve this manuscript, by responding to your comments we have been able to improve the clarity of the text and to provide more background for broad readership on how the different techniques work and on the various parameters that we are measuring in this study.

Main Issues:

Comment 2: The background of CellTiter-Glo should be briefly explained in the manuscript, facilitating the understanding of the broad readers. Also, in the Introduction, the CTG abbreviation should be elaborated, which is missing in the current version.

Response 2:

Thank you for noting this. To clarify what Cell Titer-Glo (CTG) is, we added a description to paragraph 2 of the introduction to briefly introduce CTG as an important metabolic assay for measuring cell viability. This addition also includes the definition of the CTG abbreviation which we mistakenly omitted in the submitted draft as well as an explanation of an 'endpoint assay'. We also added a short discussion at the beginning of the paragraph in methods section about CTG to include more details about what CTG measures.

Comment 3: The sentences in some places are challenging to understand. It is recommended to reorganize them to make the descriptive sentences more well-thought-out.

Response 3: We have gone through the manuscript and made a number of changes to improve clarity. In addition to smaller clarifications throughout the manuscript (highlighted in blue text in the revised manuscript), some of the more notable edits include:

Second paragraph in the section *Measurement of specific growth rate from QPI data*:

First, the rate of mass accumulation, or cell growth rate (dm/dt), can be used to characterize cell growth. In healthy cells, the growth rate is constant as cells accumulate mass during

each cell cycle (DMSO control, green line in **Fig. 1g**, **Fig. S3g,j**, **Movie M4-6**). The **cell growth** rate is typically proportional to the mass of the cell or cluster.

Second paragraph in section *Determination of sensitivity, EC₅₀, and depth of response (DoR)*:

For conditions with a response, the DoR is computed from the fitted Hill curve as the difference between the asymptotes at the highest and lowest concentrations, normalized by the asymptote at low concentration. This normalization accounts for differences in the control growth rates of each cell line. The DoR is used to determine how toxic a therapy is to a particular cell line (**Fig. 2b**).

Second paragraph in section *Measurement of heterogeneity and tracking of outliers*:

There was little relationship between the measured heterogeneity during treatment (SD at EC₅₀) and EC₅₀ or ToR, indicating that the impact of drugs on growth heterogeneity provides a measurement of drug response that is independent of sensitivity and speed of response (**Fig. 4g**, **Fig. S13**).

First paragraph in **Discussion**:

QPI predictions of both which drugs a given population of cells will not respond to and the concentration at which cells demonstrate sensitivity to therapy (EC₅₀) are strongly concordant with traditional CTG measurements. Additionally, QPI offers several additional metrics for characterization of drug response at the single cell level

Comment 4: This paper involves many meaningful QPI parameters (SGR, EC₅₀, DoR, ToR at EC₅₀, and SD at EC₅₀). It is recommended to list them in a table and give a detailed description to facilitate the reader's understanding and practice.

Response 4: This is a great idea that will improve clarity for the reader. We have added a new table, Table 1, in the main text that lists the QPI-derived response parameters we developed in this work. We reference this table where we first mention each parameter in the introduction (page 4, paragraph 3). We have also expanded the explanation of the calculation and physical meaning of each parameter where they are introduced in the text:

- a) In the second paragraph of the section “*Measurement of specific growth rate from QPI data*” we introduce specific growth rate, how it is related to cell growth and proliferation, as well as how it is computed.
- b) In the first paragraph of the section “*Determination of sensitivity, EC₅₀, and depth of response (DoR)*” we introduce EC₅₀ as a measure of drug sensitivity, and how it is found from fitting a Hill curve to the dose response data.
- c) In the second paragraph of the section “*Determination of sensitivity, EC₅₀, and depth of response (DoR)*” we introduce DoR as a measure of the toxicity of a therapy to a particular cell line and how it is computed using the asymptotes of the Hill curve.

- d) We added a note at the end of the section “*Measurement of time of response (ToR) from dynamic QPI data*” to introduce the notation “ToR at EC₅₀” just after the explanation of how it is computed.
- e) In first paragraph of the section titled “*Measurement of heterogeneity and tracking of outliers*” we added a description of the SD at the tested concentration nearest the EC₅₀ (SD at EC₅₀) as a relevant measure of heterogeneity in a responding cell population.

Comment 5: It will be meaningful to select drug-resistant tumor cells individually to assay various parameters of QPI, discuss whether QPI can effectively identify these cells, and discuss the heterogeneity ratio.

Response 5: Thank you for this suggestion. This is a complicated question that depends on many factors such as the cell line, control growth rate, cancer therapy/depth of response, and concentration. For example, if a fast growing cell line is treated with a drug that yields a large depth of response, QPI will be much more likely to detect an individual resistant cell than for a slower growing cell line treated with a drug yielding a small depth of response. There is, therefore, no single heterogeneity ratio that characterizes this or any other instrument using single cell growth rates to measure drug response.

To demonstrate this and estimate that relevant mixing ratio for the results shown here, we quantified the limit of QPI to identify the proportion of resistant cells in an *in-silico* mixture³³, using precision-recall analysis. Precision-recall curves have previously been shown to be an appropriate measure that is superior to receiver operating characteristics for evaluating a classifier for unbalanced classes⁴⁴⁻⁴⁶. This is case here where we are considering a fraction of resistant cells that is small relative to the size of the population. These results indicate that growth rate measurements by QPI have the ability to meaningfully detect resistant cell responses down to a mixing ratio of 0.1-2%, but also that this measurement characteristic is strongly dependent on cell control growth rate and cell drug response parameters for any given therapy.

We added a new main figure panel, **Fig. 4h** (previous **Fig. 4h** was moved to **Fig. S13**), a new supplemental figure, **Fig. S14**, with discussion in the *Measurement of heterogeneity and tracking of outliers* section of **Results**. We also added a brief discussion in the second paragraph of the **Discussion** and a new section in **Methods** (titled *In silico mixing analysis*) describing how we performed this analysis.

Thank you very much for your time, and for the insightful comments that you gave us on our manuscript. We found your feedback to be very helpful and to have substantially improved the quality of the manuscript.

REVIEWERS' COMMENTS:

Reviewer #1 (Remarks to the Author):

The revised manuscript has been improved. In my opinion now the manuscript is suitable for publication in the present form.

Reviewer #2 (Remarks to the Author):

The author has addressed my concerns, and the existing version has been greatly improved. It is recommended to accept it for publication.

Reviewer #1:

The revised manuscript has been improved. In my opinion now the manuscript is suitable for publication in the present form.

Response 1:

We would like to thank Reviewer #1 for their input throughout the review process.

Reviewer #2:

The author has addressed my concerns, and the existing version has been greatly improved. It is recommended to accept it for publication.

Response 2:

We would like to thank Reviewer #2 for their input throughout the review process.